# Recent Advances in Exosomal miRNA Biosensing for Liquid Biopsy

**DOI:** 10.3390/molecules27217145

**Published:** 2022-10-22

**Authors:** Bingqian Lin, Jinting Jiang, Jingxuan Jia, Xiang Zhou

**Affiliations:** 1College of Chemistry and Molecular Sciences, Wuhan University, Wuhan 430072, China; 2College of Life Sciences, Wuhan University, Wuhan 430072, China

**Keywords:** exosomal miRNA, biosensing, liquid biopsy

## Abstract

As a noninvasive detection technique, liquid biopsy plays a valuable role in cancer diagnosis, disease monitoring, and prognostic assessment. In liquid biopsies, exosomes are considered among the potential biomarkers because they are important bioinformation carriers for intercellular communication. Exosomes transport miRNAs and, thus, play an important role in the regulation of cell growth and function; therefore, detection of cancer cell-derived exosomal miRNAs (exo-miRNAs) gives effective information in liquid biopsy. The development of sensitive, convenient, and reliable exo-miRNA assays will provide new perspectives for medical diagnosis. This review presents different designs and detection strategies of recent exo-miRNA assays in terms of signal transduction and amplification, as well as signal detection. In addition, this review outlines the current attempts at bioassay methods in liquid biopsies. Lastly, the challenges and prospects of exosome bioassays are also considered.

## 1. Introduction

Cancer is one of the major diseases that threaten human health today, and its incidence and mortality rates are on a continuous rise [1]. Clinical research and practices have shown that the development of precision diagnosis and treatment technologies to achieve early diagnosis and early treatment of tumors is an important strategy to reduce the mortality rate and improve the survival rate of cancer patients [2]. As the gold standard technology for tumor diagnosis, tissue biopsy can obtain pathological information of tumors, including molecular biological characteristics, which provides important reference values for tumor diagnosis, as well as treatment [3]. However, traditional tissue biopsy techniques require invasive sampling procedures or even surgical assistance, and they have limitations such as large sampling bias, difficulty in sampling due to deteriorating clinical conditions or deep tumor location, and sampling lag [4]. Therefore, it is of great significance to explore noninvasive strategies for accurate early tumor diagnosis and monitor. Unlike noninvasive imaging for cancer monitoring [5,6], liquid biopsy is a technique that detects, analyzes, and monitors tumors by analyzing various body fluid samples such as blood or urine [7]. Compared with traditional tissue biopsy techniques that are limited to tumor progression at a single timepoint and suffer from bias due to tumor heterogeneity, liquid biopsy is one of the frontier hotspots in tumor diagnosis because it is less invasive, is convenient for multiple sampling, and can monitor tumor progression in real time.

At present, the main targets of liquid biopsy include circulating tumor cells (CTCs), circulating tumor DNA (ctDNA), and extracellular vesicles, such as exosomes [8]. Among these, exosomes derived from cancer cells may serve as a promising biomarker for several cancers [9]. Exosomes, a type of extracellular vesicle (EV), 40–160 nm in size, are secreted by all eukaryotic cells [10]. They have stable structures consisting of bilateral phospholipid membranes with proteins, DNA, mRNAs, microRNAs (miRNAs), etc. inside. These cargoes are proven to play a vital role in cell-to-cell communication [11]. Special markers associated with tumor-derived exosomes may be enriched, which may be useful for diagnosis. In particular, unique nucleic acids contained in exosomes are potentially reliable biomarkers in the diagnosis and progress monitoring of cancer. MicroRNAs are a major class of single-stranded, noncoding RNA with a length of 19–25 nucleotides (nts), which are essential regulators of gene expression, especially in cancer [12]. Recent studies have identified miRNAs in exosomes as potential biomarkers for liquid biopsy and are expected to be used in future clinical tests.

In this review, we briefly introduce the potential of exosomal miRNAs (exo-miRNAs) as biomarkers in liquid biopsy and present recent advances in biosensors, as well as the attempts of bioassays for exo-miRNAs in clinical applications. In contrast to Wu’s review [13], this paper systematically classifies the latest advances in detecting exosome-derived miRNAs with a focus on applications in liquid biopsy.

## 2. The Potential of Exo-miRNAs as Biomarkers in Liquid Biopsy

### 2.1. MicroRNAs as a Biomarker

Growing evidence has shown that miRNAs play an important role in the process of cancer development, including tumorigenesis, metastasis, and treatment resistance [14]. In most cases, miRNAs bind to their complementary sequences in the 3′ untranslated region (UTR) of target mRNAs to modulate their target [12]. In addition to the predominant mechanism, other non-canonical mechanisms have been demonstrated. Massive research has revealed that miRNAs are greatly involved in human health by regulating more than 60% of protein-coding genes and account for about 1% of the genome [15]. Additionally, the altered expression levels of miRNAs are associated with chromosomal abnormalities in their parent cells. Since then, a large number of studies have made efforts to investigate the noncoding transcriptomes from various cancer types, trying to provide appropriate miRNA candidates as a biomarker in liquid biopsy [16]. Comprehensive studies of the dysregulated profile of miRNAs in circulating environments such as blood have shown that they are associated with cancer [17]. Thus, altered levels of relevant miRNAs in liquid biopsy may provide rich information in cancer diagnosis.

### 2.2. Exo-miRNAs as Biomarker in Liquid Biopsy

In general, the release of miRNAs circulating in the body fluids follows two paths. The passive path mainly relies on tissue damage, apoptosis, and necrotic cell death. On the other hand, cells can actively encapsulate miRNAs in extracellular vesicles, such as exosomes and ectosomes. It has been shown that about 10% of secretory miRNAs are enriched in exosomes, while the remaining are associated with proteins in the circulation [18,19]. Taking advantage of the membrane structure, the miRNAs show improved stability against the degradation of RNases. Moreover, the intact structure of exosomes is not affected by non-physiological conditions, such as repeated freezing and thawing, extreme pH, or long-term storage, thus allowing the internal miRNAs to remain stable, laying the foundation for the sensitivity of miRNA detection [20].

Recent studies have demonstrated that exosomes derived from cancer cells are ideal candidate biomarkers for early cancer diagnosis and therapeutic monitoring. It has been reported that miRNAs are involved in the pathogenesis of various diseases, including cancer, through the exosomes that are taken up by the recipient cells as cargo [21,22]. Increasing evidence has revealed that miRNAs transferred by exosomes contain valuable information about the original cell types, the recipient cells, and disease progression. It has been reported that the exo-miRNAs derived from cancer cells can promote cell proliferation, migration, and angiogenesis [20], such as exosomal-miR-21 (exo-miR-21) [23], exo-miR-23a [24], exo-miR-100 [25], and so on [26,27]. For instance, exo-miR-21 was investigated as a promising biomarker for breast cancer [28] and ovarian cancer [29]. So far, 2838 miRNAs have been reported to be encapsulated in exosomes from several kinds of cell types (Exocarta) [30]. Therefore, it is of great significance to use exo-miRNAs as a detection target in liquid biopsy for early diagnosis, progression monitoring, and therapy evaluation of cancer.

Several methods for exo-miRNAs analysis have been reported. The common methods used to quantitative and profile miRNAs are quantitative reverse transcription real-time PCR (qRT-PCR) [23], digital PCR [31], and next-generation sequencing (NGS) [32]. However, despite the superior performance of these methods, they have some shortcomings in the application of liquid biopsy. Quantitative real-time PCR (qRT-PCR) has been recognized as the gold standard for miRNA analysis due to its excellent sensitivity and flexibility. However, this method has some problems such as non-absolute quantification, false positives, reliance on expensive equipment, and a large number of biological samples, which limit its application in routine clinical practice [31]. In addition, the ddPCR and NGS usually need high costs and tedious operations, limiting the use of these assays in a large number of clinical liquid biopsy scenarios [33]. Many biosensors are designed to explore disease-associated markers, such as detecting cancer-associated markers for cancer diagnosis [34,35,36]. Compared with the above methods, miRNA detection using biosensors does not require complex steps with shortened detection time and does not require large and expensive instruments, meeting the requirements of rapid and high sensitivity for clinical detection in liquid biopsy.

## 3. Signal Transduction and Amplification Strategies in Exo-miRNA Biosensors

Many methods of signal transduction and amplification are commonly applied in the detection of miRNAs. Due to the nucleic acid nature of miRNAs, various nucleic acid-based strategies and nucleic acid-synergized strategies are required in the process of capture and signal transduction and amplification. In this section, we introduce signal transduction and amplification methods of exo-miRNA detection.

### 3.1. Nucleic Acid-Based Strategies

#### 3.1.1. Various Nucleic Acid Capture Probes


Individual capture probes


Molecular beacons (MBs) are the most commonly used probes for the detection of exo-miRNAs. MBs are designed as a stem-loop structure where the loops are complementary to the target molecule, while the sequences at each end complement each other to form a hairpin structure. One end of the MB is labeled with a fluorescent dye and the other end with a quencher group. In the natural state, the MB forms a hairpin shape, which leads to the quenching of the fluorescent dye. In the presence of the target, the complementary pairing of MB with the target opens the stem-loop structure, which restores fluorescence. According to this design, MBs were used to detect exo-miRNAs from different sources, such as urine and blood. Several MBs were designed for detection of exo-miRNAs, such as miR-21 [37,38], miR-375 [39], miR-574-3p [38,39], and so on [40,41]. For example, Zhou et al. utilized MB to specially detect miRNA-146a inside a captured extracellular vesicle [42]. In addition, signal amplification can be performed using an enzyme-assisted method. Zheng et al. used end-modified 2′-O-methyl MB (eMB) that is resistant to enzymatic degradation of DNase I [43]. It is only degraded upon binding to the target, which releases the target to bind to the next MB, forming a signal amplification reaction with a detection limit of 2.5 pM.

In addition to natural DNA probes, several nucleic acid analogs have been synthesized by various modified backbones for improving thermal stability and discrimination ability. Among these, locked nucleic acid (LNA) probes are representative analogs with restricted sugar motifs. In LNA, the “locked” 2′-O, 4′-C-methylene-linked ribonucleotides showed a high affinity with complementary RNA [44]. To improve the sensitivity of exo-miRNA detection, Lee et al. presented a work that utilized two LNA probes to hybridize with miRNA simultaneously to form a sandwich structure [45]. The sandwich structure was only formed in the presence of the perfectly matched miRNA, thus enabling the ability of precise discrimination of a single base. In addition, the hybridization had enhanced affinity due to the adjusted Tm value. The LNA capture probes were modified on gold nanopillars and LNA detection probe were Cy3-labeled for strong SERS signals with a detection limit down to 1 aM.

Along with LNA, peptide nucleic acid (PNA) is another commonly used nucleic acid analog with polypeptide backbones [46]. The uncharged PNA backbone shows high affinity for its complementary DNA or RNA molecules, due to the lack of electrostatic repulsion in the base-pairing interactions between PNA and the natural nucleic acid. Taking the advantage of PNA, Xiao et al. demonstrated a PNA-functionalized nanochannel for tumor-derived exo-miRNA sensing with high sensitivity and specificity [47]. In this assay, PNA was covalently modified on the surface of nanochannel for enhanced hybridization efficiency with target miRNA, due to the duplex of PNA/miRNA changing the charge density on the channel surface with increasing ion current. This biosensor achieved high sensitivity of 75 aM detection limit without a signal amplification process due to the neutrality of PNA, the high surface area, and the nanoscale confinement effect of the nanochannels.


Assembled capture probes


Due to the intramolecular or intermolecular nature of Watson–Crick base-pairing reactions, DNA can be programmed to form delicate nanostructures. Take the advantage of predictability and programmability, various design rules and assembly methods have been used for constructing complex DNA nanostructures. Through precise spatial control, capture probes and different reporter groups such as fluorescence and nanoparticles can be immobilized at designed positions to enable detection reactions, as well as signal output.

Luo et al. developed a LNA modified Y-shaped structure capture probe for exo-miR-21,which is shown in Figure 1A [48]. The methylene blue-modified Y1 probe (MB-Y1) and ferrocene-modified Y2 probe (Fc-Y2) formed a Y-shaped structure on an electrode. In the presence of miR-21, the miRNA hybridized with Fc-Y2, thus triggering the strand displacement reaction on the Y-shaped structure, leaving the free MB-Y1 to form a hairpin structure. During this process, the signals of MB and Fc changed respectively, thus achieving the detection of miRNA with a detection limit of 2.3 fM.

Tetrahedral DNA nanostructures (TDNs) represent an outstanding DNA nanoarchitecture that has rigid structures with precise dimensions. This specialty allows TDNs to be immobilized on the surfaces of the sensor carrier with a fixed direction, thus making a programmable “soft lithography” for electrochemical sensing [49]. Subsequently, Liu et al. reported an “assembly before testing” strategy using TDNs as capture probes for exo-miRNA testing, as shown in Figure 1B [50]. In the presence of miRNA, the TDNs hybridized with partial targets and the other part of the target complement to the PNA which were modified on an electrode. The immobilized TDNs could adsorb massive electroactive molecules, thus resulting in a high sensitivity of 34 aM. Another advantage of the assembled DNA structure is that the signal probes can be specifically fixed within a defined space, thereby increasing the local concentration of the probes and improving the efficiency of the reaction. As shown in Figure 1C, Xu et al. applied a Janus wireframe DNA cube for accelerated signal amplification reaction after target recognition [51]. In this method, different hairpins were integrated into the assembly of the Janus wireframe for catalytic hairpin assembly (CHA). In the presence of the target, the hairpins on the Janus wireframe DNA cube were opened in boundary and cycles for restored fluorescent signals. The engineered signal probes on the DNA structures greatly reduced the reaction time with a detection limit of picomole level.

**Figure 1 molecules-27-07145-f001:**
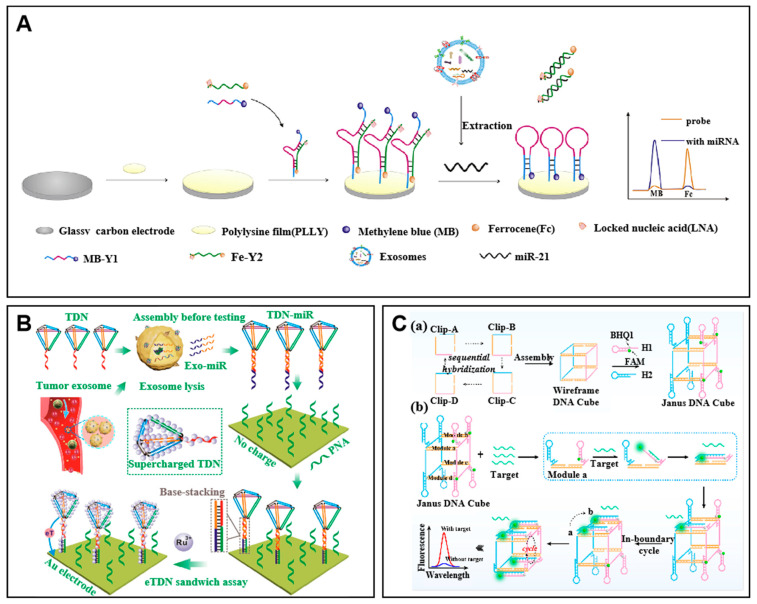
Examples of assembled DNA capture probes. (**A**) Schematic of the Y-shaped structure-based ratiometric electrochemical biosensor for the detection of exosomal miR-21. Adapted with permission from [48]. Copyright 2019 Elsevier. (**B**) Schematic of the tetrahedral DNA nanostructure (TDN)-based DNA capture probes for target exo-miRNA capturing. After formation of the TDN–miR complex, the PNA hybridizes with the complex as a sandwich constructed on a gold electrode for further sensing. Adapted with permission from [50]. Copyright 2021 American Chemical Society. (**C**) Schematic of the fluorescent sensor for detection of exo-miRNA based on Janus wireframe DNA cube. (a) The procedure of Janus wireframe DNA cube assembly. (b) The reaction of the 3D nanomachine. Adapted with permission from [51]. Copyright 2022 Elsevier.

#### 3.1.2. Signal Amplification by DNA Self-Assembly

To obtain high sensitivity, signal amplification methods are essential in biosensors. The signal amplification reaction of DNA self-assembly generally does not require the involvement of enzymes, thus making the reaction relatively simple. The hybridization chain reaction (HCR) is an effective isothermal signal amplification method based on DNA self-assembly that has been widely used [52]. The HCR system contains two nicked hairpin DNAs and a single-stranded initiator DNA. Initially, the initiator DNA opens hairpin 1 (H1), followed by the opening of hairpin 2 (H2) by H1; then, the H2 can further open H1, resulting in the cyclic hairpin opening step to form a long double-stranded structure similar to alternating copolymers [52]. The designed hairpins allow exponential assembly, thus enabling the assembly to generate a large number of signals in biosensing. The HCR can be modified for enhanced signal amplification performance. Guo et al. reported an ultrasensitive assay based on HCR for the detection of exo-miRNA (Figure 2A) [53]. The gold electrode was modified with hairpin DNA (hpDNA). In the presence of miR-122, the hpDNA opened and triggered the HCR to form long nicked double helices which can capture massive RuHex (electrochemical signal reporting molecules) for increasing signals, resulting in a detection limit of 49 aM. In addition, several efforts have been made for improving the performance of HCR. Kim et al. applied the HCR system on a hydrogel structure, which reduced the sample volume with estimated detection limits of 1–10 amol and 10–100 amol for miR-6090 and miR-3665, respectively [54].

The catalytic hairpin assembly (CHA) reaction is another commonly used signal amplification method by DNA self-assembly. A typical CHA reaction system consists of two DNA strands that can form a hairpin structure [55]. In the initial state, the two DNA strands themselves form a stable hairpin structure due to complementary interactions, such that no interaction occurs. When the trigger strand is added, one DNA strand is opened due to the toe structure, and the opened strand can open the other strand, which then forms the thermodynamically most favorable double-stranded structure. Through the reaction of the trigger strand, a large number of two hairpin structures open to form a double strand. In order to improve the sensitivity and specificity of exo-miRNA detection, Zhang et al. designed a biosensor based on a step polymerization hairpin assembly (SP-CHA) system for the detection and analysis of exosomal miR-181 (Figure 2B) [56]. The SP-CHA system consists of three catalytic hairpins (H1, H2, and H3), an electrode surface capture probe, and target miRNA. MicroRNA-181 in the sample was captured by a capture probe on the electrode surface, and the first hairpin structure H1 was opened by a strand displacement reaction, which turned on the formation of a three-way T-shaped assembly intermediate. These T-shaped assembly intermediates acted like seeds and began to grow on their own, eventually forming tandem products of different lengths. The catalytic hairpins H1 and H3 were modified with biotin. The finally obtained biotinylated tandem product could react with streptavidin-alkaline phosphatase (ST-AP). AP could catalyze the conversion of substrate α-NP into irreversible electroactive products, generating highly sensitive electrochemical signals with a detection limit of 7.94 fM.

The assembled DNA nanostructures are widely in conjugation with signal amplification methods of DNA assembly. For example, Zhang et al. developed a rapid detection platform based on multifunctional TDN-assisted catalytic hairpin assay (CHA) [57]. The basic components of the system were two TDNs named T1 and T2 assembled by one-step annealing and modified with CHA and free DNA sequences on adjacent tetrahedral vertices. The target miRNA can trigger the CHA on T1 and lead to the capture of T1 on the electrode. Then, the free strands on T1 can trigger the CHA reaction on T2, thus resulting in the capture of T2 on the electrode. Similarly, free strands on T2 continued to initiate CHA on T1, leading to the exponential amplification by plentiful capture of T1 and T2 on the electrode surface, followed by binding of RuHex for generating electrochemical signal. The proposed method can realize ultrasensitive detection of exo-miRNA with a detection limit of 7.2 aM within 30 min.

Toehold-mediated strand displacement reaction (TSDR) is another enzyme-free DNA assembly reaction. The toehold (single-stranded DNA fragments) can bind to the invader strand, triggering branching migration at rates that vary by more than six orders of magnitude [58]. As shown in Figure 2C, Liu et al. developed an enzyme-free analysis of exo-miRNA based on TSDR [59]. The developed assay consists of a localized DNA cascade displacement reaction (L-DCDR) and assembled methylene-binding DNA nanosheets (MB-DSN).

**Figure 2 molecules-27-07145-f002:**
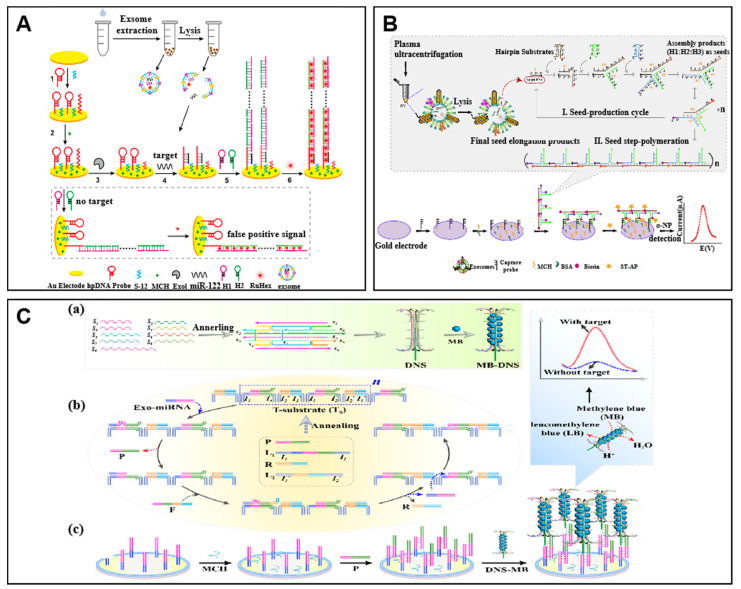
Examples of signal amplification by DNA assembly. (**A**) Schematic of hybridization chain reaction (HCR) based electrochemical biosensor for the detection of exo-miRNA. Adapted with permission from [53]. Copyright 2020 American Chemical Society. (**B**) Schematic of the electrochemical biosensor based on a step polymerization catalytic hairpin assembly (SP-CHA) assay. Adapted with permission from [56]. Copyright 2021 Elsevier. (**C**) Schematic of the electrochemical biosensing assay based on localized DNA cascade displacement reaction (L-DCDR). (a) The procedure of MB-DNS assembly. (b) The procedure of L-DCDR. (c) The binding of the MB-DNS on the electrode. Adapted with permission from [59]. Copyright 2020 American Chemical Society.

In the presence of target miRNA, the target triggered the TSDR by binding to the toehold 1 (T1) strand and releasing the P strand, thus exposing the pre-locked toehold 2 (T2) strand. Then, the F strand binds to the exposed T2 strand, thus replacing the target and the R strand by strand displacement. The released target triggered the next circle of the reaction and released massive P strands. The massive P strands hybridized with the capture probe on the electrode surface and MB-DNS simultaneously, thus leading to amplified electrochemical signals of a detection limit down to 64 aM. Similarly, Xia et al. used strand displacement assay coupled with carbon dots and acridone derivate to construct a ratiometric fluorescent bioprobe sensor for the detection of exo-miR-21 [60]. The carbon dot-modified DNA strand and acridone derivate-modified DNA strand were displaced successively, thus resulting in the change of the FRET signal for ratiometric fluorescent detection. With the incorporation of LNA into the DNA strand, the signal-to-noise ratio was improved with a detection limit of 3.0 fM.

#### 3.1.3. Enzyme-Assisted Isothermal Amplification

Unlike the signal amplification method based on DNA assembly, the enzyme-assisted method can trigger the amplification reaction of DNA at a constant temperature with the assistance of different isothermal amplification enzymes, thus achieving exponential signal amplification. Rolling circle amplification (RCA) is a frequently used tool that possesses circular DNA as a template to form a long single-stranded DNA or RNA sequence [61]. In this method, the template is used as a guide sequence for cycling, resulting in long linear DNA or RNA sequence with multiple tandem repeats in the presence of specific DNA or RNA polymerases and primers. These tandem repeats can be functional sequences or combined with complementary sequences with motifs. Sulaiman et al. developed a one-pot method for extracellular vesicle (EV) derived miRNA (EV-miRNA) detection, which is completed with EV lysis and miRNA capture inside encoded hydrogel microparticles (Figure 3A) [62]. The captured target miRNA is ligated to a universal linker that is complementary to the circular template amplified by RCA. The obtained product is ligated with biotinylated reporters and streptavidin labeled PE (SA-PE) to obtain a substantial fluorescent signal for a low limit of detection to zeptomole and a large dynamic range.

In addition to DNA polymerases, nucleases are applied in the exo-miRNA biosensors. The clustered regularly interspaced short palindromic repeats/CRISPR associated nuclease (CRISPR/Cas) toolbox has been widely investigated as an efficient and versatile genetic engineering tool in various fields [64]. CRISPR/Cas9 is one of the widely used nucleases in Cas proteins that can perform precisely double-stranded (dsDNA) cleavage with RNA guidance. Wang et al. combined the RCA with CRISPR/Cas technologies to build an isothermal amplification platform with multitarget detection [65]. A padlock probe was utilized in RCA for single-base identification assisted by HiFi Taq DNA ligase. Then, the long linear DNA with massive tandem repeats was hybridized with a large amount of TaqMan probes to form PAM structures containing dsDNA which can be recognized by Cas9. The fluorescent intensity increased through the cleavage of Cas9 of formed target dsDNA with the detection limit of 90 fM for EV-derived miRNA-21. By introducing multicolor tags on TaqMan probes, the platform can achieve multiplex detection of various targets (miRNA-21, miRNA-221, and miRNA-222).

Exponential amplification reaction (EXPAR) is another enzyme-assisted isothermal amplification method with high efficiency. The amplification template used in EXPAR consists of two concatenated sequences that are complement to the target DNA. The target DNA is introduced to the system as a primer to be extended by the DNA polymerase. The resulting duplex contains the recognition site of a nicking enzyme that can be cleaved to release the newly synthesized oligonucleotide. The resulting oligonucleotide, thus, acts as a new primer for the next cycle of the reaction. The cyclic repetition of the process leads to the exponential amplification of the target as high as 108-fold in a few minutes. Because of the advantages of high amplification efficiency and high detection sensitivity, EXPAR has been integrated with electrochemical technology for the detection of various analytes in wide fields. Qian et al. used a quantitative assay based on EXPAR for the detection of exo-miRNAs (Figure 3B) [63]. After on-chip capture and lysis of exosomes, the miRNAs extracted from exosomes were quantified by EXPAR to achieve a detection limit down to 100 fM.

### 3.2. Nucleic Acid-Synergized Strategies

#### 3.2.1. Nucleic Acid Synergized Nanomaterials

Materials consisting of basic units with dimensions between 1 and 100 nm in at least one dimension are called nanomaterials, including metallic, inorganic, and organic nanomaterials. They exhibit expressive advantages of flexible and modular structures, large surface area, small size, and good biocompatibility [66]. In addition, some materials have shown properties suitable for biological sensing, such as enzyme-mimicking features, fluorescent properties, and superparamagnetic behavior. In biosensing assays, nanomaterials can be used as capture carriers, signal transduction, or amplification modules.

Gold nanoparticles (AuNPs) are widely used in biosensing systems as carriers due to their easy synthesis and high nucleic acid ligand adsorption properties [67]. Carbon dots are widely used as signal molecules due to their excellent photoluminescence properties and biocompatibility [68]. Using these two nanomaterials, a fluorescent biosensor based on gold nanoparticles combined with a DNAzyme locator was proposed to simultaneously detect breast cancer (BC)-associated exo-miRNAs (Figure 4A) [69]. In this work, the gold nanosurface was modified with activity-inhibited DNAzymes and a high density of carbon dot (CD)-labeled substrates. In the absence of the target miRNAs, the DNAzymes were inactive due to their hairpin structure, and the fluorescence of CDs was quenched by AuNPs. In contrast, the presence of target miRNAs can open the hairpin of the DNAzymes, allowing them to hybridize with the substrate, thereby cleaving the substrates and releasing CDs for fluorescence restoration. The DNAzyme walking process was accomplished by triggering subsequent substrate reactions as programmed. In this process, a small number of target miRNAs can cause cleavage of a large number of CD-labeled substrates on the AuNP surface, thus presenting an amplified fluorescence signal. The simultaneous analysis of the two exo-miRNAs can be achieved by modifying CD substrates with different luminescence. The results showed that the biosensor could quantitatively detect exo-miRNAs in the linear range of 50 fM to 10 nM. In addition, the system has potential in clinical serum samples. Similarly, Liu et al. used AuNPs as carriers for the loading of massive DNA probes called spherical nucleic acid (SNA) [28]. The targets bind to the electroactive tags adsorbed SNA- and PNA-modified gold electrodes simultaneously for electrochemical sensing, resulting in a detection limit down to 49 aM.

Carbon nanotubes (CNTs) are ideal channel materials for field-effect transistors (FETs) for their sensitivity change upon adsorbing various chemical and biological molecules. Li et al. developed a label-free FET biosensor on a CNT for the detection of exo-miRNAs (Figure 4B) [70]. Unlike conventional CNT FET biosensors, the used CNTs were modified with a Y_2_O_3_ insulating layer with deposited AuNPs. In the presence of target miRNA 21 (exo-miR-21), the anchored DNA capture probe on the AuNPs binds to the target exo-miR-21, thus resulting in p-type electrostatic doping. The detection concentrations of exo-miR-21 ranging from 1 aM to 1 nM indicate the high sensitivity of this method.

Metal–organic frameworks (MOF) is a class of crystalline porous materials with a periodic network structure formed by interconnecting metal nodes with linkers of organic ligands through self-assembly. It has both the rigidity of inorganic materials and the flexibility characteristics of organic materials, thus presenting excellent characteristics such as high porosity, low density, large specific surface area, regular pores, adjustable pores, and topological diversity. As shown in Figure 4C, Li et al. developed an electrochemical biosensor using a target-triggered cascade primer exchange reaction (PER) combined with the MOF@Pt@MOF nanozyme. In this method, the target triggered the PER and produce a long DNA strand, thus exposing the capture probe for binding to the nanozyme. The used nanozyme was constructed by two layers of MIL-88 with platinum nanoparticles (PtNPs) in the middle layer. The nanozyme showed excellent peroxidase-like activity in the H_2_O_2_ catalytic reaction. The single-stranded DNAs were modified on the surface of the nanozyme for acting as a signal probe, which generated an amplified electrochemical signal for a detection limit of 0.29 fM [71].

The inherently small size of nanomaterials gives them an advantage for application to miniaturized biosensors, thus promising small, low-cost, portable, and user-friendly medical diagnostic point-of-care or bedside diagnostic tools. In addition, the high surface area properties of nanomaterials allow them to be used directly as signal capture carriers for the application of isolating and capturing disease biomarkers in the circulatory system. The ability of nanostructured materials to directly engage with the sensing environment (electrolytes, markers, etc.) accelerates signal transduction, which in turn improves the robustness and sensitivity of the analysis and reduces detection limits. By modifying nanomaterials, they can be applied in label-free assays for direct detection of biological systems.

#### 3.2.2. Nucleic Acid Synergized Microfluidics

Microfluidic devices allow sample process and analysis in a miniaturized platform which is known as “lab-on-a-chip” [72]. Facilitated by the designed microscale size of architectures, microfluidic devices provide precise fluidic control in nano- or microlevel. Defined geometries of microfluidic chips enable the reactions between buffers and reagents, as well as analytes, via flow forces. Benefiting from miniaturization, microfluidics has unique advantages in processing speed and costs with small amounts of analyte. Microfluidics can be deployed as DNA synergized substrates for efficient and sensitive exo-miRNA capture and sensing. In this section, we discuss recent advances in DNA-synergized microfluidic analysis of exo-miRNAs.

Sung et al. designed an integrated chip for automated miRNA detection which contains two modules: the sample treatment module and miRNA detection module [29]. In the sample treatment module, the EVs are isolated by anti-CD63 coated magnetic beads, followed by the extraction of EV encapsulated miRNA-21 by complementary DNA modified magnetic beads. The quantification of miRNA-21 by digital PCR (dPCR) was performed in the miRNA detection module with a detection limit of 11 copies per mL. The further verification of clinical samples confirmed the excellent applicability of the proposed chip in liquid biopsy. In addition to the signal output of the PCR method, Ramshani et al. developed an integrated chip with chemical-free EV lysis and membrane sensing without PCR amplification, achieving one-step absolute quantification of EV-derived miRNAs [73].

Unlike general microfluidic chips, test strips are a type of paper-based microfluidic commonly used in point-of-care testing. Kim et al. utilized a lateral flow assay (LFA) assisted with DNA barcode sequences which can be captured by immobilized DNA probes on LFA for multiplex detection of exo-miRNAs [74]. In this assay, designed capture DNA probes were immobilized on the nitrocellulose membrane of the LFA. The target miRNAs were captured by a template with a stem-loop RT primer, resulting in nicked DNA sequences with stem-loop and target miRNA sequences. After that, the DNA sequences simultaneously bound with biotin-labeled reverse primer and DNA barcode containing forward primer for asymmetric-PCR. The resulting products were captured by the DNA probe on LFA and detected by streptavidin-coated gold nanoparticles, achieving a sensitivity of 95.24% and a specificity of 100.0%.

## 4. Signal Sensing Strategies in Exo-miRNA Biosensors

### 4.1. Surface-Enhanced Raman Scattering (SERS)

Detection of exo-miRNAs mainly relies on fluorescent sensors derived from capture probes or signal amplification products. However, in the complex environment of exosomes, the fluorescence-based detection method has some problems, such as fluorescence signal self-quenching and fluorescence background signal interference. This results in the reduced sensitivity and detection limit of the method, making it difficult to distinguish low levels of target miRNAs. Surface-enhanced Raman scattering (SERS) is a surface sensitive technology of scattering enhanced by using nanostructures such as molecules or plasmonic magnetic silica nanotubes adsorbed on rough metal surfaces [75]. This technique has the characteristics of high sensitivity and low background noise. Therefore, this strategy can reduce the interference caused by bioenvironmental components in exo-miRNA detection and increase the possibility of practical application.

The commonly used SERS signal tag is the combination of Raman reactive dye and metal nanoparticles. When the signal tag is applied to the detection of exo-miRNA in actual samples, the complex biological environment causes instability on the surface of the tag and pollution of the signal tag caused by reducing agents or surfactants in the environment. To solve this problem, Ma et al. developed a new SERS analysis strategy by combining stable SERS reporter elements and duplex-specific nuclease (DSN)-assisted signal amplification [76]. The SERS reporter element includes three important elements: signal reporter element, recognition element, and separation element. They attached Rhodamine 6G (R6G) to AuNPs and then encapsulated them in AgAu alloy shell nanoparticles to form the signal reporter element ARANPs. The DNA capture probes (CP) were designed to be completely complementary to the target exo-miRNA. The streptavidin-coated silica microbead (SiMB) was used as an effective separation element. ARANPs and SiMB were covalently linked to both ends of CP by Au–S and biotin–streptavidin binding. Therefore, CP could couple ARANPs to SiMB surface to construct a SERS switch. In addition, DSN was used to promote exo-miRNA signal amplification. After the target exo-miRNA binds to CP, the hybrid duplex formed by them can be specifically cleaved by DSN, leading to the release of SERS signal (ARANPs) on the surface of SiMB. Meanwhile, the removed target exo-miRNA remains intact and participates in the next round of target binding for cyclic amplification. After cycling, a large number of ARANPs were released to generate strong SERS signal. Therefore, the SERS intensity of the solution was directly related to the concentration of target miRNA with a detection limit of 5 fM.

The optical biosensors based on SERS often use precious metals or nanoparticles to analyze a single bioanalyte adsorbed on a rough metal nanostructure. However, when the concentration of target bioassays is extremely low, its strong background signal limits the application of SERS in complex clinical samples. Nevertheless, 3D plasmonic nanostructures can provide a higher SERS effect. The SERS active sites that are significantly amplified by the incident electric field in this structure are called electromagnetic “hotspots”. The enhancement effect can identify the Raman spectral fingerprint of a single molecule. It overcomes the problems of low sensitivity and high background signal when a traditional Raman signal is used to identify multiple analytes in a complex environment.

Kim et al. combined 3D plasmonic nanostructures with SERS strategy for designing a quantitative label-free biosensor to detect urinary exo-miRNA [77]. As shown in Figure 5A, they coupled self-assembled DNA probes to three-dimensional gold nanostructured particles called SAP-AuNPs. In the presence of target miRNAs, gold nanoparticles with DNA probes complementary to the target miRNA and adjacent gold nanostigma generated a large number of three-position plasma hotspots by self-assembling plasma coupling effect. Thus, extremely high SERS signal amplification was induced to enhance the sensitivity of the biosensor and reduce its detection limit to avoid strong background signals. The SERS biosensor could successfully identify changes in the expression level of exo-miRNA. The detection of exo-miRNAs in urine samples provides a promising approach to differentiate prostate cancer (PC) patients from healthy patients.

### 4.2. Surface Plasmon Resonance (SPR)

Surface plasmon resonance (SPR) is an optical sensing technology that is highly sensitive to the refractive index of surface media [78]. The SPR biosensing technology is produced by combining SPR with biological immunoassay. Although the emergence of SPR biosensing technology has significantly improved the sensitivity of isolating exosome populations, its expensive instrumentation and time-consuming procedures have limited its clinical application. Song et al. combined DNA self-assembly technology with a plasma sensor system to achieve accurate detection of exosome-miRNAs biomarkers in the serum of clinical Alzheimer’s patients (Figure 5B) [79]. They designed and synthesized a rod-shaped version of DAPA. It consists of three gold nanoparticles formed by DNA self-assembly with a 2 nm distance between the two gaps in the middle, which can induce plasmonic coupling effects. In addition, they used nucleic acid (LNA) probes to distinguish the target miRNAs in exosomes from other highly homologous miRNAs. Thus, the detection limit is very low, and the performance of the biosensor is improved.

### 4.3. Electrochemical Detection

In recent years, electrochemical-based methods have attracted significant interest, mainly because of their inherent high sensitivity and specificity, short reaction time, simple operation steps, and wide dynamic range. In addition, the electrochemical-based approach is well suited to miniaturization for portable testing, which facilitates the realization of large-scale multiple electrochemical devices [66]. The development of electrochemical biosensors provides a very effective strategy to achieve rapid, economical, and amplification-free miRNA detection.

Boriachek et al. proposed an amplitude-free electrochemical miRNA detection method using the superparamagnetic gold-loaded nanoporous iron oxide nanomaterials in electrochemical sensors [80]. The detection of exo-miRNAs in complex biological samples was realized. The biotinylated probe captured the target miRNA and hybridized the captured probe with commercially available streptavidin magnetic beads. After hybridization, the target miRNA was released from the probe by thermal action. The released target miRNA was adsorbed onto the SPE-AU via RNA–gold affinity interactions. In the presence of the [Fe(CN)6]^4^^−/3−^ redox system, differential pulse voltammetry (DPV) was used for monitoring.

Cheng et al. subjected the catalyzed hairpin assembly (CHA) to streptavidin–biotin interactions to assist in the channeling of the silver nanoparticle aggregation strategy [81]. An enzyme-free electrochemical biosensor with double signal amplification was developed to achieve ultrahigh sensitivity detection of exosomes in biological samples. They immobilized the catalytic hairpin structure HP1 on the gold electrode. The target miRNA-21 in the sample could open the HP1 structure. When the HP1 structure was opened, a new viscous end was exposed, which could open the free catalytic hairpin structure HP2. At the same time, the target miRNA-21 was replaced. The structure of HP1 was opened to expose a new sticky end. This sticky end could open the free catalytic hairpin structure HP2 and displace the target miRNA-21 at the same time. The displaced target miRNA could also continue to hybridize with other HP1 to achieve signal amplification. In addition, HP2 bound to HP1 was also modified with biotin. Through biotin–streptavidin interaction, silver nanoparticles were deposited and aggregated on the gold electrode, thus achieving the purpose of electrical signal amplification. This double signal amplification strategy greatly improves the specificity and sensitivity of exosome detection in biological samples and reduces the detection limit without enzyme.

In order to simplify the steps of miRNA detection and reduce the operation time, Saha et al. applied competitive hybridization assays to electrochemical sensors (Figure 5C) [82]. They used a two-step competitive hybridization assay to rapidly detect EV-miRNAs in real clinical samples. They developed a layered sensor with a high surface area. The electrode is a millimeter-sized star-shaped electrode made of gold nanoparticles. They attached a single-stranded DNA probe that captures the target miRNA to the star-shaped electrode. Then, the unlabeled target miRNA was captured with the capture probe. The unreacted capture probe was then hybridized to the signal DNA barcoding. Such barcodes are single-stranded DNA with the same sequence as the unlabeled target, and their ends are also modified with a redox species (MB). The output signal of the barcode is inversely proportional to the concentration of the target miRNA; a lower target concentration results in a higher output signal. This design can reasonably convert the low microRNA concentration into a large electrochemical signal, thus achieving rapid and ultrasensitive detection of miRNAs in complex clinical samples.

**Figure 5 molecules-27-07145-f005:**
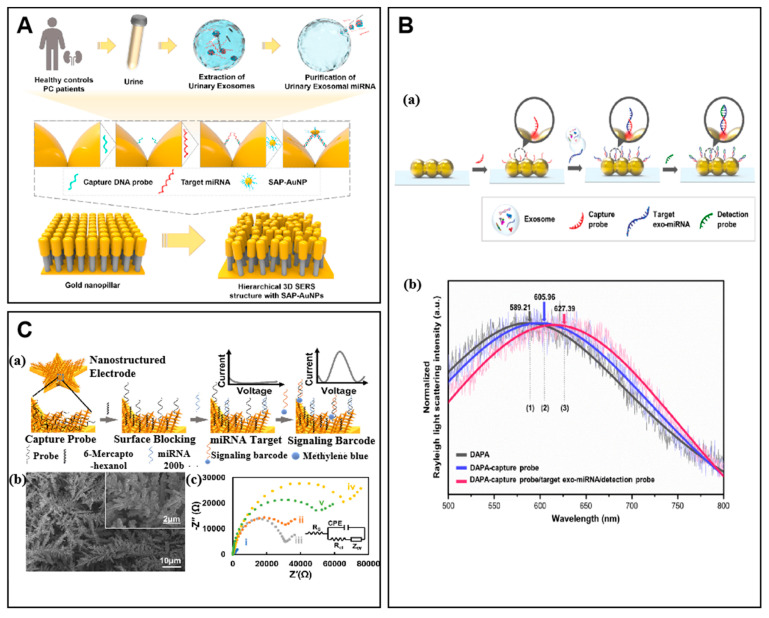
Examples of signal sensing strategies in exo-miRNA biosensors. (**A**) Schematic of the hierarchical 3D SERS structure-based exo-miRNA detection. Reprinted with permission from [77]. Copyright 2022 Elsevier. (**B**) The label free plasmonic biosensor for exo-miRNA detection. (a) The schematic illustration of DNA-assembled advanced plasmonic architecture (DAPA)-based plasmonic biosensor to accurately detect exo-miRNA. (b) The representative Rayleigh light scattering spectra. Adapted with permission from [79]. Copyright 2021 Elsevier. (**C**) The two-step competitive hybridization assay for miRNA detection from extracellular vesicles. (a) Schematic of the two-step competitive hybridization assay for cancer-related miRNA detection. (b) SEM image of the star-shaped electrodes. (c) The Nyquist curves obtained from electrochemical impedance spectroscopy for the nanostructured Au electrode. Adapted with permission from [82]. Copyright 2021 American Chemical Society.

## 5. In Situ Detection of Exo-miRNA

Most of the methods used for exo-miRNA detection require exosome isolation and miRNA extraction. The complex process may cause loss or contamination of miRNAs. In situ detection of miRNA in exosomes avoids losses during extraction, while allowing the reaction to be restricted to exosomes, thus improving the efficiency and sensitivity of detection.

### 5.1. Exo-miRNA Detection by Membrane Fusion

One of the commonly used in situ detection methods involves the membrane fusion of exosomes. In nature, delivery of biomaterials can be achieved through membrane fusion, thus being important in biological functions. By using membrane fusion, the detection of miRNA in exosomes can be achieved without disrupting the exosomal membrane, thus ensuring the integrity of nucleic acids. In addition, encapsulation of the signal molecule by another intact membrane structure allows confinement of the reaction within the membrane and increases the local concentration of the signal molecule, thus increasing the sensitivity of the reaction and reducing the limit of detection.

Gao et al. developed a rapid and efficient strategy for the detection of exo-miRNA based on membrane fusion between exosomes and a virus-mimicking fusogenic vesicle (Vir-FV) [83]. In this strategy, the Vir-EV was genetically engineered with fusogenic proteins from human parainfluenza virus with MB encapsulated. The fusogenic proteins initiated the fusion of Vir-EV with exosomes, followed by the hybridization of MB with target exo-miRNA, thus achieving a large fluorescent signal. The signals from tumor exosomes and normal exosomes can be distinguished by the flow cytometry for individual exosomes. The whole reaction process needed no pre-isolation of human serum, thus achieving a rapid detection time within 2 h. This platform has made great progress in improving the efficiency of vesicle fusion. Similarly, Zhou et al. combined strategy of membrane fusion using Vir-FV with microfluidic chip for ultrasensitive exo-miRNA detection [84] without requiring RNA extraction.

In addition to genetic engineering membranes, vesicles made from exosome-derived cell membranes were used according to the same phenotypic features. Cao et al. designed a homotypic recognition method to detect exo-miRNAs based on catalytic DNA machinery enveloped by biomimetic vesicles which fuse with cancer exosomes (Figure 6A) [85]. In this method, two hairpin probes, electroactive MB-labeled H1 and N3-labeled H2 are camouflaged by specific biomimetic vesicles which are immobilized on anti-CD44@IMBs. They are fused with their homologous cancer exosomes to initiate catalytic DNA machinery by endogenous miRNA generating hybridized hairpin DNA. Then, lysis and magnetic separation are performed to gather hybridized DNA onto a DBCO-modified gold electrode to trigger electrochemical signals. Through this method, a large linear range of exosome concentration can be detected quantitively with the lower limit of 518 particles·mL^−1^ with different terms of cancer exosomes, making it possible for clinical cancer therapy.

In contrast to membranes made of natural biological sources, Yang et al. reported an immuno-biochip using cationic lipoplex (CLP) fusion with exosomes [37]. The CLPs were encapsulated with RNA targeting molecular beacons (MBs). When the exosomes were captured by antibodies on the immuno-biochip, the positively charged CLP fused with negatively charged exosomes and led to the opening of the MBs; thus, the restored fluorescent intensity could be detected by total internal reflection fluorescence (TIRF) microscopy.

### 5.2. Exo-miRNA Detection by Penetration

The nanoparticles and defined DNA nanostructures can be used to penetrate into exosomes for the detection of miRNAs. Gold nanoparticles are commonly used as carriers to penetrate into the lipid bilayer of the exosomes. Zhao et al. reported a nanoflare-assisted thermophoretic sensor (TSN) for highly sensitive detection of exo-miRNAs [87]. The nanoflare was constructed by modifying antisense single-stranded DNA of miRNA on AuNPs, and the DNAs were prehybridized with Cy5 modified short complementary sequences. The Cy5 groups on nanoflares were quenched in proximity to the AuNPs. The nanoflares can penetrate into the exosomes and replace the Cy5 labeled short sequences with target miRNA hybridization, thus leading to the restoration of the fluorescence intensity detected by a thermophoretic sensor. For investigation of exosome uptake of AuNPs, Shi et al. used different sizes of DNA-functionalized AuNPs for uptake efficiency of exosomes and sensing sensitivity of exo-miRNAs [88]. The results showed that smaller AuNPs were more easily taken up by exosomes and yielded higher sensitivity.

In addition to nanoparticles, Chen et al. used DNA nanotubes for spatial confinement-derived cascade amplification for in situ sensing of exo-miR-21 [86]. As shown in Figure 6B, a double-accelerated DNA cascade amplifier (named DDCA) was constructed by a DNA nanocube with two hairpin DNAs (designated H1 and H2) modified with Cy3 and Cy5, respectively. The H1 and H2 were assembled into the DNA nanocube by base-pairing. When the DDCA entered the exosomes, the target miRNA bound to the H1 and triggered the CHA reaction within the nanotube, thus resulting in the fluorescence resonance energy transfer (FRET) between Cy5 and Cy3. In this assay, the reaction was defined within the nanoreactor space, largely accelerating the reaction speed, with a detection limit of 9.8 × 10^4^ particles/μL.

### 5.3. Exo-miRNA Detection In-Situ

Jiang et al. developed a double SERS biosensor, which can simply, accurately, and quickly separate exosomes from serum samples and directly detect specific miRNAs in exosomes in situ [89]. The Raman signal reporter molecule 5,5′-dithiobis-(2-nitrobenzoic acid) (DTNB) was modified on gold nanoparticles, called Au@DTNB. Then, the locked nucleic acid (LNA) was modified on the Au@DTNB surface to form SERS tag. This SERS tag can be transferred to exosomes by incubation, which has been confirmed by many studies [90]. When target miRNAs are combined with complementary LNA sequences with SERS tags, SERS tags can assemble and induce numerous particles to trigger hotspot aggregation, thus generating electromagnetic hotspots that can significantly provide enhanced SERS signals. Next, exosomes were separated because titanium oxide can bind phosphate groups with high specificity. The Fe_3_O_4_@TiO_2_ core–shell nanoparticles were used to separate the SERS-tagged exosomes. Then, the external magnet was used to capture it, and the Raman laser was used for SERS detection. Therefore, the synergistic effect of hotspot assembly in exosomes and exosome segregation and aggregation leads to a dual increase in Raman signaling, thereby improving the sensitivity of detecting exo-miRNAs. This method is the first to apply the SERS method of target-triggered hotspots to the detection of exo-miRNAs. Its simple operation and high sensitivity make it a promising tool for clinical diagnosis of exo-miRNAs in situ.

## 6. Exo-miRNA Biosensing for Liquid Biopsy

Using the methods mentioned above, many studies have tested the accuracy and sensitivity of the methods in actual clinical samples; relevant cases usually include breast cancer [28], liver cancer [41,73], colorectal cancer [74], ovarian cancer [29], and gastric cancer [41]. Typically, these are small samples of clinical attempts. In breast cancer, Liu et al. examined the exo-miR-21 levels of six patients and six healthy volunteers which could be well discriminated [28]. Kim et al. examined exo-miR-92a and exo-miR-141 for the detection of colorectal cancer patients with a 100% specificity [74]. Sung et al. found that the level of miRNA-21 in ovarian cancer patients was higher than noncancer and endometrial cancer patients [29]. Ramshani et al. identified the overexpression of miR-21 from liver cancer patients [73]. In contrast to clinical testing attempts that look solely at miRNA concentrations, the method Cao et al. developed can distinguish between phenotypes [85]. The ER-positive luminal subtype of breast cancer patients was analyzed with good discrimination.

However, the sample sizes used in existing testing cohorts are far from the large sample sizes used in clinical testing, and the standardization of their methodological assays remains to be investigated. In addition, most of the current assay attempts are for exo-miRNA detection in the blood (plasma or serum), with urine accounting for a minority [42], along with a few for exo-miRNA detection in cerebrospinal fluid [91]. Therefore, there is a need to develop multiple assays to study the association of miRNAs in different body fluids with diseases.

There are still some limitations in applying exo-miRNA biosensors to liquid biopsy:(1)Variation on sample processing methods. Most of the methods used for biosensors require exosome extraction from body fluids followed by miRNA extraction; thus, the pretreatment process including exosome and miRNA extraction from body fluids can affect their stability and quantitative levels. Moreover, the strict control of RNase-free environment during the operation can affect the final results.(2)Detection inconsistency due to assay variations. In clinical applications, assay variation among various assays and between laboratories becomes a problem. The signal amplification methods applied in the above methods and the signal detection methods, including laboratory operations and detection techniques, can cause differences in exo-miRNA quantitation between different platforms, limiting their application to some extent.(3)Batch stability and reproducibility of biosensors in complex samples. In clinical samples, it is often necessary to face a complex biological matrix containing massive unrelated biological macromolecules, such as proteins and nontarget nucleic acids. These substances may adsorb nonspecifically to the sensor surface, leading to unstable signal fluctuations that affect reproducibility.

## 7. Summary and Perspectives

In liquid biopsy, the design of corresponding bioanalytical methods for specific biomarkers allows for rapid and accurate diagnosis. Combined with the advantage of noninvasiveness, liquid biopsy can provide information related to disease diagnosis, disease process monitoring, and prognostic feedback for the development of disease. With the growing recognition that miRNAs play an important role in cancer progression and have specific characteristics corresponding to different types of cancer, researchers have studied different types of miRNAs to examine their potential as biomarkers. Tumor cells encapsulate some of the miRNAs in EVs or exosomes through active processing. Since EVs or exosomes have a bimolecular lipid layer structure, they can protect miRNAs from degradation and, thus, maintain their stability. The detection of miRNAs derived from EVs or exosomes will be an important direction for liquid biopsy. In this review, we discussed recent several different signal transduction and amplification methods including nucleic acid-based strategies and nucleic acid-synergized strategies. We also described signal detection methods including SERS, SPR, and electrochemical detection in exo-miRNA biosensing. In addition, we outlined attempts at exo-miRNA biosensors in clinical applications. Despite the potential of exo-miRNA biosensor applications, many issues deserve further refinement.


The standardization of exo-miRNA detection


To date, most biosensors have been mere proof-of-concept demonstrations and have been attempted with a small number of clinical samples. Experimental conditions and assay standards vary from laboratory to laboratory. Moreover, there is a lack of standardized procedures for exosome collection, exosome degradation, and miRNA extraction and preservation. In addition, an in-depth study with validation of miRNAs corresponding to different cancer tests is also needed to form the corresponding guidelines. The use of bioinformatics can assist the miRNA expression profile for further research on exo-miRNA biosensing.


The extension of sample sources


Most articles analyzed EV cargoes from serum and plasma. However, EVs are present in various human body fluids, such as saliva, tears, and urine. Appropriate information can be obtained by studying EVs or exosomes from nontraditional samples. Therefore, the extension of liquid biopsy to nontraditional sample sources requires further research and validation for application to clinical testing as an aid to diagnosis.


In situ exo-miRNA detection for single exosome


In situ detection of miRNAs in exosomes is based on the nondestructive detection of exo-miRNAs, thus avoiding the extraction step of miRNAs. In situ investigation of exo-miRNAs provides intact information about miRNAs in single exosomes. In situ detection studies of exo-miRNAs by developing assays with temporal and spatial control can reduce background signals and improve detection accuracy.

Although the biosensing detection of exo-miRNAs has made rapid progress, further exploration is needed to obtain more efficient, fast, and cost-effective methods and find more precise ways to characterize the function of exo-miRNAs in body fluids, thus providing effective strategies for cancer prevention, early identification, and therapy in the near future.

## Figures and Tables

**Figure 3 molecules-27-07145-f003:**
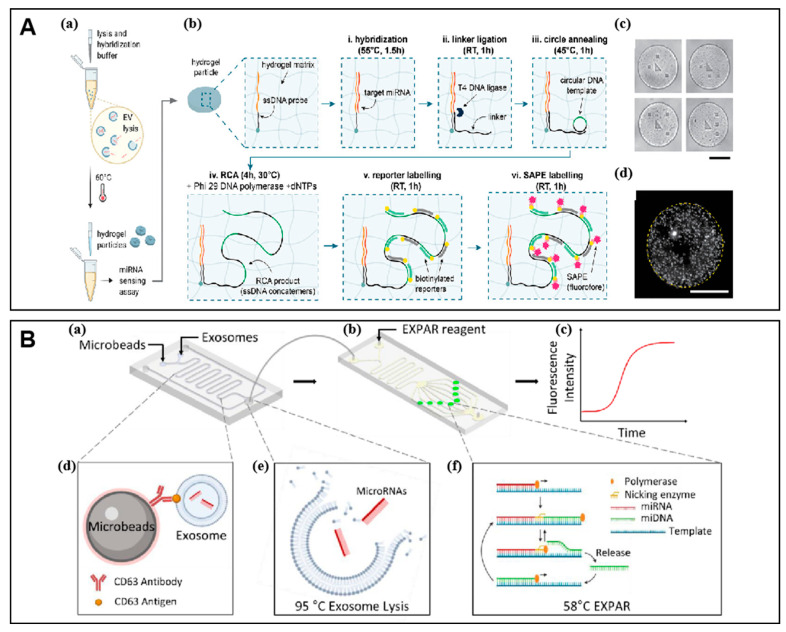
Examples of enzyme-assisted signal amplification for exo-miRNA detection. (**A**) Schematic of the miRNA capture and detection by encoded hydrogel microparticles. (a) The one-pot procedures of EV lysis and miRNA capture. (b) RCA in hydrogel microparticles. (c) Microscopic images of multiple sensing with four graphically encoded hydrogel particles. (d) The single hydrogel microparticle imaged by confocal fluorescence microscopy. Adapted with permission from [62]. Copyright 2022 Wiley-VCH GmbH. (**B**) Schematic of the on-chip exo-miRNA analysis. (a–c) The exosomes were captured and lysis for detection of miRNA by isothermal EXPAR assay. (d) The capture of CD-63 expressed exosomes by anti-CD63-coated magnetic beads. (e) Release of exosomal miRNAs by 95 °C lysis. (f) The isothermal EXPAR assay. Adapted with permission from [63]. Copyright 2021 Elsevier.

**Figure 4 molecules-27-07145-f004:**
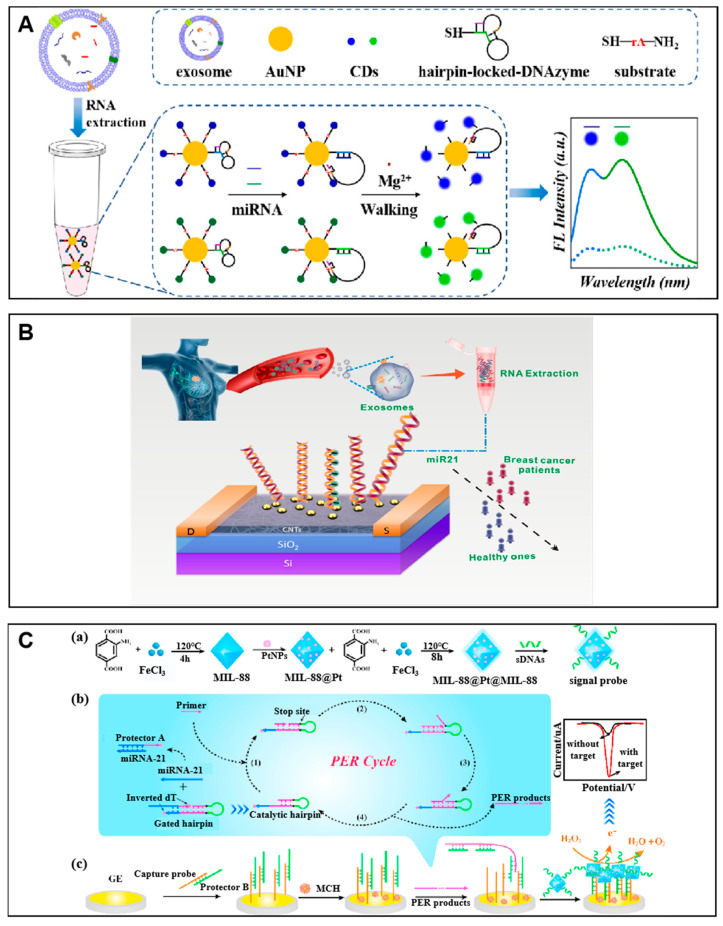
Examples of DNA synergized nanomaterials for exo-miRNA detection. (**A**) Schematic of the AuNP@CD inorganic nanoflare-DNAzyme biosensor. Reprinted with permission from [69]. Copyright 2019 Elsevier. (**B**) Schematic of the DNA functionalized carbon nanotube FET biosensor for ultrasensitive detection of exosomal miR-21. Reprinted with permission from [70]. Copyright 2021 American Chemical Society. (**C**) Schematic of the MOF@Pt@MOF nanozyme-assisted electrochemical biosensor for exo-miRNA detection. (a) The synthesis steps of the MOF@Pt@MOF. (b) The mechanism of the PER cycle. (c) The representative electrochemical signal generated by MOF@Pt@MOF. Adapted with permission from [71]. Copyright 2020 Elsevier.

**Figure 6 molecules-27-07145-f006:**
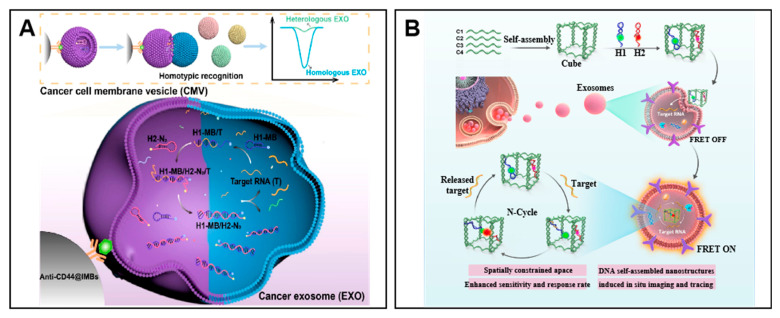
Examples of in situ detection of Exo-miRNA. (**A**) Schematic of homotypic recognition-driven membrane fusion for exo-miRNA detection. Reprinted with permission from [85]. Copyright 2022 American Chemical Society. (**B**) Schematic of the assembled DNA cube machine for penetration of the exosomes. Adapted with permission from [86]. Copyright 2022 American Chemical Society.

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
