# Peer review of "Recent Advances in Exosomal miRNA Biosensing for Liquid Biopsy"

_molecules, 2022, doi:10.3390/molecules27217145_

Round 1

Reviewer 1 Report

The manuscript entitled “Recent Advances on Exosomal miRNA Biosensing for Liquid 1 Biopsy” by Lin et al., aims to explore advancement on exosomal miRNA biosensing for liquid biopsy. The manuscript is well written supported by illustrations. It advances our knowledge in line of detection strategies of recent exosomal miRNA assays in terms of signal transduction and amplification. However, my concerns are as below:

1.     The text of figure not legible. Please increase the font size.

2.     Is there any limitations of miRNA biosensing for liquid biopsy? Please discuss in the text.

Author Response

Thank you for your valuable suggestions. We have added a description of the limitations of miRNA biosensing for liquid biopsy in section “6. Exo-miRNA Biosensing for Liquid Biopsy”. And the texts of the figures were adjusted for clarity.

Reviewer 2 Report

The introduction section needs to add some information from latest finding on cancer as well as biosensors. I recommend following articles to explore and cite accordingly

MEMS based cantilever biosensors for cancer detection using potential bio-markers present in VOCs: a survey

Gosmann, Dario, et al. "Promise and challenges of clinical non-invasive T-cell tracking in the era of cancer immunotherapy." EJNMMI research 12.1 (2022): 1-14.

Micro-cantilevered MEMS Biosensor for Detection of Malaria Protozoan Parasites

Huang, Weicai, Yuming Jiang, Wenjun Xiong, Zepang Sun, Chuanli Chen, Qingyu Yuan, Kangneng Zhou et al. "Noninvasive imaging of the tumor immune microenvironment correlates with response to immunotherapy in gastric cancer." Nature communications 13, no. 1 (2022): 1-14.

2. The quality of figures should be immensely improved as texts on figures become legible

3. The conclusion needs to be rewritten.

4. there are many grammatical errors, and authors may take the help of professional editing services.

The article is technically good and can be published with minor changes as suggested.

Author Response

Thank you for your valuable suggestions. We have carefully checked the grammar of the manuscript and corrected it accordingly. All changes are made in revision mode. We have added some references from latest finding on cancer as well as biosensors into the introduction part in the revised manuscript as recommended. And the conclusion part has been rewritten. In addition, the texts of the figures were adjusted for clarity.